# Rosai–Dorfman Disease between Proliferation and Neoplasia

**DOI:** 10.3390/cancers14215271

**Published:** 2022-10-27

**Authors:** Ismail Elbaz Younes, Lubomir Sokol, Ling Zhang

**Affiliations:** 1Department of Pathology, Cleveland Clinic, Cleveland, OH 44195, USA; 2Department of Hematology and Oncology, Moffitt Cancer Center, Tampa, FL 33612, USA; 3Department of Pathology, Moffitt Cancer Center, Tampa, FL 33612, USA

**Keywords:** Rosai–Dorfman disease, sinus histiocytosis, MAPK pathway, gene mutation, histiocytic disorder

## Abstract

**Simple Summary:**

Rosai–Dorfman disease (RDD) was a benign histiocytic proliferative disorder rather than a neoplastic process. Emergent molecular studies have shown recurrent somatic gain-of-function mutations in genes of the MAPK pathway (e.g., *NRAS*, *KRAS*, *MAP2K1*, and *ARAF*) in a subset of RDD, suggesting a clonal histiocytic proliferative process. This review encompasses clinical characteristics, updated subclassification, diagnostic approaches, and treatment strategies, with an emphasis on the molecular profiles of RDD. This study includes the latest international consensus on diagnosing and managing this disease. This review will provide novel insights on the latest discoveries in RDD.

**Abstract:**

Rosai–Dorfman disease (RDD) is a rare myeloproliferative disorder of histiocytes with a broad spectrum of clinical manifestations and peculiar morphologic features (accumulation of histiocytes with emperipolesis). Typically, the patient with RDD shows bilateral painless, massive cervical lymphadenopathy associated with B symptoms. Approximately 43% of patients presented with extranodal involvement. According to the 2016 revised histiocytosis classification, RDD belongs to the R group, including familial and sporadic form (classical nodal, extranodal, unclassified, or RDD associated with neoplasia or immune disease). Sporadic RDD is often self-limited. Most RDD needs only local therapies. Nevertheless, a small subpopulation of patients may be refractory to conventional therapy and die of the disease. Recent studies consider RDD a clonal neoplastic process, as approximately 1/3 of these patients harbor gene mutations involving the MAPK/ERK pathway, e.g., *NRAS*, *KRAS*, *MAP2K1*, and, rarely, the *BRAF* mutation. In addition to typical histiocytic markers (S100/fascin/CD68/CD163, etc.), recent studies show that the histiocytes in RDD also express BCL-1 and OCT2, which might be important in pathogenesis. Additionally, the heterozygous germline mutation involving the FAS gene *TNFRSF6* is identified in some RDD patients with an autoimmune lymphoproliferative syndrome type Ia. *SLC29A3* germline mutation is associated with familial or Faisalabad histiocytosis and H syndrome.

## 1. Introduction

Rosai–Dorfman disease (RDD) was initially described by Pierre-Paul Destombes, who reported four cases in 1965 [1]; afterward, Reed and Azoury outlined and reported a classic case of RDD [2]. Later, Rosai and Dorfman described 34 cases and coined the name sinus histiocytosis with massive lymphadenopathy, which was later changed to RDD [3,4,5]. The largest cohort collected 423 cases from an international registry in 1990 where both nodal and extranodal RDD were documented [5].

RDD predominantly occurs in young black children and clinically manifests with massive bilateral cervical lymphadenopathy. Extranodal and cutaneous forms are present. Clinical course is variable, ranging from self-limited process to disseminated refractory disease with increased associated mortality. Given the disease heterogenicity, treatment options are different for RDD, including local or systemic approaches. Histologically, a biopsy of the lesion reveals characteristic features of abnormal S100^+^, CD68^+^, and CD1a^−^ histiocytes with infrequent to overt emperipolesis. RDD can be an isolated or combined disorder. Increased polyclonal plasma cells and fibrosis in the background are often associated with IgG4 lymphoproliferative disorder. It is also not uncommon for RDD to be associated with autoimmune diseases, malignant tumors, and rare hereditary disorders [6].

The etiology of RDD is unclear. It has been postulated that an infectious agent is the etiologic cause of RDD; nonetheless, no microorganism has been detected. Epstein–Barr virus (EBV) was thought to play a role in RDD because many patients with RDD were positive for EBV; however, in situ hybridizations for EBV were negative [7]. Other virus candidates (e.g., HHV-6, HIV, and cytomegalovirus) were also proposed but never approved.

Recent studies showed that approximately 1/3 of RDD these patients harbor gene mutations involving the MAPK/ERK pathway, e.g., *NRAS*, *KRAS*, *MAP2K*1, and, rarely, *BRAF*, indicating a neoplastic process rather than a reactive disorder.

Genetic predisposition or hereditary forms of RDD has been hypothesized, as cases have been described in twins or family members, which supports this hypothesis [8]. Germline mutations in *SLC29A3* (at 10q23), which is associated with H syndrome (histiocytosis-lymphadenopathy plus syndrome), Faisalabad histiocytosis, and pigmented hypertrichotic dermatosis with insulin-dependent diabetes, have been found in cases of familial RDD. Another germline mutation (*TNFRSF*) that is found in autoimmune lymphoproliferative syndrome (ALPS) type I is also found in RDD.

This review outlines clinicopathologic features; updates the subclassification and treatment of RDD; emphasizes the novel molecular findings behind this entity; and aims to guide clinicians and pathologists on how to appropriately reach the diagnosis and proceed with targeted therapy.

## 2. Discussion

RDD is an abnormal proliferation of histiocytes with varieties of clinical pictures, either isolated or with other diseases, which requires an integrated clinical, radiological, pathological, and molecular diagnostic approach. According to the Society of Histiocytes expert consensus in 2016, histiocytosis and neoplasms of the macrophage-dendritic cell lineage are reclassified into five groups: (1) Langerhans-related (L group); (2) cutaneous and mucocutaneous (C group); (3) malignant histiocytosis (M group); (4) RDD (R group); and (5) hemophagocytic lymphohistiocytosis and macrophage-activation syndrome (H group) [9]. Despite RDD being considered an independent entity distinguished from other histiocytoses, the nature of RDD still needs further exploration, in particular, following the laboratory implications of the novel next generation sequencing (NGS) technique. The emergence of novel molecular data indicates that RDD is a neoplastic process. RDD was recently listed in the 5th Edition of the World Health Organization (WHO) Classification of Myeloid and Histiocytic/Dendritic Neoplasms [10].

## 3. Epidemiology

RDD is a relatively rare entity, with an incidence of 1:200,000, with around 100 new cases diagnosed annually in the United States [6]. RDD usually affects African American children and young adults, with a slight male predominance. The mean age at diagnosis of RDD is approximately 20.6 years with a male-to-female ratio of 3:2 [5]. In contrast to classic or nodal RDD, cutaneous RDD on the other hand is a different entity, with a mean age of 45 years and more common incidence in Caucasians and Asians [11].

## 4. Subclassification

Under the R group of non-Langerhans cell histiocytosis (LCH), RDD can be further classified into five subgroups: classical (nodal), familial, extranodal, neoplasia associated, and immune disease-associated RDD [9].

The classical RDD subgroup includes IgG4 lymphoproliferative disorder or those without IgG4 syndrome. Upon involved sites, extranodal RDD is further subgrouped into bone RDD, central nervous system (CNS) RDD (with or without IgG4 syndrome), single-organ, or disseminated RDD. The single-organ RDD does not include lymph nodes or CNS RDD and is further divided into two subtypes: with or without IgG4 syndrome. Of note, cutaneous RDD has been reclassified to be included in the C group under the nonxanthogranuloma family [9].

Certain hereditary conditions predispose to RDD. Familial RDD is unique, which can be linked with H syndrome or ALPS [9], and is considered in a separate category that will be discussed under the session of molecular mechanisms.

RDD has been associated with variable kinds of neoplasia or immune diseases. Approximately 10% of RDD coexists with immunologic diseases: Systemic lupus erythematosus (SLE) has been reported with RDD in many case reports. They are different from the pathogenesis perspective; however, some emerging case reports show that some mutations may lead to a newly discovered disease entity called RAS-associated autoimmune leukoproliferative disease, which is caused by gain-of-function mutations in the RAS-family (*NRAS* and *KRAS*). A case report shows the p.G13C mutation in the *KRAS* gene in a patient presenting with SLE and RDD, rendering the diagnosis of RAS-associated autoimmune leukoproliferative disease [12]. Idiopathic juvenile arthritis and autoimmune hemolytic anemia have been reported in association with RDD [13].

RDD can occur in patients with a history of lymphomas. However, composite lymphomas and RDD rarely occur. Many types of lymphomas can co-occur with RDD, including classic Hodgkin lymphoma and non-Hodgkin lymphoma. The two most common types of lymphomas associated with RDD are follicular lymphoma and nodular lymphocyte-predominant Hodgkin lymphoma. The latter was changed to nodular lymphocyte-predominant B-cell lymphoma, according to the most recent International Consensus Classification of Mature Lymphoid Neoplasms [14]. Additionally, RDD can co-occur with other lymphomas (marginal-zone lymphoma, peripheral T-cell lymphoma), cutaneous clear-cell sarcoma, myelodysplastic syndromes (MDS), allogeneic stem transplant for precursor B-cell acute lymphoblastic leukemia, and concurrent with or following L-group histiocytosis, e.g., LCH, Erdheim Chester disease (ECD), or malignant histiocytosis [6,15,16].

## 5. Clinical Presentation

RDD usually presents with bilateral cervical lymphadenopathy. Mediastinal, groin, and, rarely, or retroperitoneal cavity can be involved. Patients with RDD are often accompanied with B symptoms (fever, night sweat, and weight loss). Around 43% of patients develop extranodal disease [13], most commonly involving skin, nasal cavity, and orbit. Salivary, spleen, and testes can be involved [8]. Bone RDD manifested with solitary or multifocal lytic lesions, involving in long bones, vertebrae, and sacrum. [16,17]. The organ specific clinical presentation and imaging findings are summarized in Table 1.

Laboratory evaluation of RDD patients shows approximately 2/3 of patients present with normochromic normocytic anemia, leukocytosis (neutrophilia most commonly). Poly-clonal hypergammaglobulinemia was reported in approximately 90% of patients; increased erythrocyte sedimentation rate in approximately 90%; and hypoalbuminemia in approximately 60%. The CD4:CD8 ratio was found to be decreased. Other laboratory findings are present, such as elevated ferritin level and autoimmune hemolytic anemia. 

According to the international expert consensus at the 32nd Histiocyte Society Meeting in 2018 and National Comprehensive Cancer Network (NCCN) Guidelines for Histiocytic Neoplasms (2020), comprehensive systemic physical examination should be conducted, including head and neck, intrathoracic/pulmonary/cardiovascular, gastrointestinal (GI), renal, genitourinary (GU) system, neuroendocrine, CNS, and cutaneous symptoms. The following recommendations are used for baseline evaluation of new/suspected cases of RDD [6,23]. 

Medical history
Constitutional symptoms;Organ affection (head, eyes, ears, nose, and throat; cardiovascular; pulmonary; GI; GU; skin; CNS; and endocrine);History of autoimmune disease, LCH, or other histiocytic lesions, as well as hematologic malignancies;Family history for children.
Physical examination
Lymphadenopathy;Organomegaly;Cutaneous and extranodal lesions;Neurologic changes.
Radiological evaluation
All patients should have whole-body positron emission tomography (PET)/computed tomography (CT);Selected patients should have CT sinuses with contrast, high-resolution CT chest, magnetic resonance imaging of orbit/brain with contrast and magnetic resonance imaging of spine with contrast;Selected patients for organ-specific ultrasound.
Laboratory evaluation
Complete blood cell count, complete metabolic panel, erythrocyte sedimentation rate;Serum immunoglobulins;Coagulation studies, C-reactive protein, uric acid, and lactate dehydrogenase (LDH);Hemolysis panel (Coombs, haptoglobin, reticulocytes, and blood smear);Panel for autoimmune diseases (ALPS panel, antinuclear antibody, rheumatoid factor, HLA B27);NGS targeted gene mutations in RAF-RAS-MEK-ERK pathway;If familial form is suspected, then NGS test for *SLC29A3* (germline mutations);Bone-marrow biopsy (if cytopenia or abnormal peripheral blood smear are present);Lumbar puncture for CNS involvement.
Subspecialty consultation as needed
Dermatology and ophthalmology evaluation before initiating MEK-inhibitor therapy.


## 6. RDD with Concurrent Disorders

Besides aforementioned lymphomas, MDS or rare solid tumors, it is not uncommon to accompany RDD with other histiocytic neoplasms and benign lymphoproliferative disorders. Herein, we only focus on the LCH, ECD, and IgG4 disorders that we will discuss here.

Diagnosis of IgG4 disease is based on recent consensus scoring made by the American College of Rheumatology and the European League Against Rheumatism [24]. Histologically IgG-4 related disease shows increased IgG4 plasma cells with a ratio of at least 0.4 IgG4: IgG plasma cells or more than 100 positive IgG4 plasma cells in a high-power field. There are five different morphologic variants (Multicentric Castleman disease–like, follicular hyperplasia, interfollicular expansion, progressive transformation of germinal centers, and inflammatory pseudotumor–like). A sixth variant has been suggested by Chen et al., yet it might be considered as an advanced case of interfollicular expansion type. Many studies have shown an increased number of IgG4 plasma cells and associated fibrosis in RDD; however, according to the criteria of the American College of Rheumatology and the European League Against Rheumatism, only from 10% to 30% of RDD concurrent IgG4-related disease [25,26,27,28]. Of note, nodal RDD tends to be slightly more affected than extranodal RDD by IgG4-related disease.

RDD can occasionally occur in patients with LCH [29,30]. There are mostly in case reports, with no case series being reported to the best of our knowledge. Immunohistochemical stains would be helpful to identify the two entities. Molecularly, LCH would often have *BRAF* V600E mutations, while RDD rarely has this kind of mutation. *MAP2K1* mutation is present in 1/3 of cases of RDD but uncommonly seen with LCH. *MAP2K1* and *BRAF* mutations in LCH are mutually exclusive. 

ECD has been reported to be often associated with LCH with coincidence rate of 15% [31] and also occur with RDD, though less frequently. A multicenter study from Pitié-Salpêtrière Hospital, Memorial Sloan Kettering Cancer Center, and the Mayo Clinic referral centers showed that 11/353 (3.1%) had overlapping RDD with ECD, which is slightly more common than other histiocytic neoplasms overlapping with RDD. None of those cases had *BRAF V600E*; however, mutations were found in *MAP2K1* [32].

## 7. Histopathology and Immunochemistry

Histologic examination shows enlarged, matted, grossly involved lymph nodes and capsular fibrosis. In lymph nodes with subtotal involvement, dilation of sinuses causes severe architectural alterations. Sinuses are obstructed by a mixed population of cells, including histiocytes, lymphocytes, plasma cells, and histiocytes. The most distinctive cells are the histiocytes; hence, the name RDD histiocytes was coined. RDD histiocytes are usually large with round-to-oval nuclei, dispersed chromatin, prominent nucleoli, and abundant clear-to-foamy or vacuolated cytoplasm. The most unique feature of these histiocytes is emperipolesis (a Greek word meaning wandering in and around). During the process of emperipolesis, the histiocytes engulf intact cells and sometimes nuclear debris and lipids. The engulfed cells remain viable and can exit histiocytes in contrast to the process of phagocytosis (Figure 1). These findings are observed in both nodal and extranodal sites; however, there is a greater degree of fibrosis and RDD histiocytes with less frequent or absent emperipolesis on extranodal sites. Extranodal RDD also appears more frequently to be fibrosis and less frequently to be histiocytosis. RDD is often associated with abundant plasma cells in the medullary cords and around the venules [8].

Fine-needle aspiration–smears and touch imprints are typically highly cellular with many histiocytes and engulfed lymphocytes (emperipolesis) against a background of mixed inflammatory cells, including plasma cells and lymphocytes. Histiocytes tend to be large with abundant cytoplasm and a round, vesicular nucleus with a small central nucleolus. In smear and imprint preparations, there tends to be a diagnostic dilemma; overlapping lymphocytes can be mistaken for emperipolesis as engulfed lymphocytes do not appear surrounded by a halo, as seen in tissue sections. In later stages of RDD, there tends to be increased plasma cells and cytoplasmic immunoglobulin inclusion (Russell bodies).

Studies have shown that RDD histiocytes express CD4, CD11c, CD14, CD68 (KP-1), and CD163 [33,34,35]. Uniquely, in RDD, the histiocytes express S100, which is a useful feature for visualizing the emperipolesis, as first described in a single case by Aoyama et al. [36] and confirmed in a larger series by Miettinen et al. [37] (see Figure 2). RDD is usually negative for pan B- or T-cell antigens, markers for Langerhans cells (CD1a and langerin/CD207), and follicular dendritic cell markers (CD21, CD23, CD35, and clusterin). RDD histiocytes are also reactive to α1-antichymotrypsin and α1-antitrypsin, which might suggest lysosomal activity.

Recent studies have shown that 1/3 of RDD cases harboring mutations in the MAPK/ERK pathway that were found to be gain-of-function mutations leading to the upregulation of p-ERK and cyclin D1/BCL-1 in the histiocytes [38,39,40]. Cyclin D1/BCL-1, a key cell-cycle regulator, represents a major downstream target of the MAPK/ERK pathway. Expression that is positive for Cyclin D1/BCL-1 can be associated with phosphorylated -ERK (p-ERK), reflecting the constitutive activation of MAPK pathway [39]. However, some cases show cyclin D1 upregulation and absence of p-ERK expression; hence, cyclin D1 might be regulated by other oncogenic mechanisms bypassing the ERK pathway [40]. Positive cyclin D1/BCL-1 staining in RDD is not associated with an underlying translocation of the *CCND1* gene. Be cautious: some reactive histiocytes also express cyclin D1/BCL-1; however, this is usually dimly expressed. The marker becomes less specific for RDD if it is weakly expressed on histiocytes [41,42]. Besides cyclin D1/bcl-1, the Mayo group study showed a subset of RDD cases also expressed p16 (64%), Factor XIIIa (30%) and phosphorylated extracellular signal-related kinase (45%) [43]. The latter two parameters appeared to be associated with multifocality of RDD [43]. As Factor XIIIa is also seen with ECD, this should be interpreted with caution.

Another marker that was frequently expressed in many RDD cases is BCL-2. Since RDD cases usually have low Ki-67 proliferation index, the expression of BCL-2 might be caused by the activation of the anti-apoptotic process [40,43]. OCT2, a unique monocyte-macrophage marker, has also been found to be expressed in most cases of RDD, in contrast to other histiocytic disorders, such as LCH and ECD [43] (Figure 2). Plasma cell markers (IgG, IgG4, kappa and lambda) are used to identify concurrent IgG4 disease (Figure 3).

Although RDD has many prominent features, which can help to distinguish this entity from other histiocytic disorders, certain features are nonspecific, e.g., emperipolesis can be identified in ECD, juvenile xanogranuloma, and malignant histiocytosis [6]. In addition to S100, histiocytic markers (Fascin, CD68, CD163, CD4, CD14) positive for RDD can be seen with numerous histiocytic disorders, especially when RDD concurs with other histiocytic lesions. Differentiation between RDD and other histiocytic disorders is sometimes challenging. For nodal RDD, differential diagnoses mainly include LCH-, ECD-, and ALK1-positive histiocytosis and infection or other malignancies associated with reactive histiocytosis.

### 7.1. Benign Disorders

Sinus histiocytosis, which is a benign nonspecific reaction in a reactive lymph node with increased histiocytes. This entity does not show emperipolesis, and few histiocytes stain for S100.Toxoplasma lymphadenitis usually present with the diagnostic histologic triad of
Sinusoidal expansion by monocytoid B cells;Follicular hyperplasia;Epithelioid histiocytes encroaching on reactive germinal centers differ from RDD, since these histiocytes usually occur in small groups with baby granulomatous changes;


Serology test and special stains would be helpful for diagnosis.

3.Hemophagocytic lymphohistiocytosis (HLH): The patients with HLH present with hemophophagocytosis that cytologically could mimic emperipolesis.

However, clinically the patients with HLH are critically ill, with disseminated disease, often life threatening, and associated with severe pancytopenia, hepatosplenomegaly, high levels of ferritin, triglycine, and soluble IL25, distinguishing it from RDD. S100 stain on phagocytic histiocytes is negative.

### 7.2. Neoplastic Disorders

LCH can be easily differentiated from RDD. Unlike RDD, Langerhans cells in LCH usually show nuclear groves and thin nuclear membranes and are often associated with abundant eosinophils and necrosis. Langerhans cells are positive for CD1a and langerin (CD207), which are negative in RDD, yet both RDD and LCH are positive for S100 [44,45]. An electron microscope shows a specific feature in cytoplasm of the LCH cells, called Birkbeck granules. *BRAF* V600E is more frequently mutated in LCH than in RDD. Another immunohistochemical stain that can be used to detect underlying *BRAF* V600E is the BRAF VE1 clone [46]. PD-L1 is more commonly positive in LCH than RDD [46].Diagnosing ECD requires a comprehensive study, including clinicopathologic, radiologic, and molecular assessment. Pathognomonic, radiologic features for diagnosis include symmetrical long-bone osteosclerotic lesion of lower limbs and sheathing of the aorta (coating of the aorta by fibrosis). Histologically, it presents with histiocytes with xanthogranulomatous changes and fibrosis against the background of inflammatory cells and Touton giant cells. Similar to RDD, the histiocytes in ECD are also positive for CD63 and CD168, a small subset positive for S-100 that could mimic RDD. However, unlike RDD, the histiocytes in ECD do not show emperiolopoiesis, and are positive for XIIIa and *BRAF* mutations (>50% of cases) [47];Classic Hodgkin lymphoma can rarely be localized in lymph-node sinuses, which can create a diagnostic dilemma in some cases; however, the presence of Reed–Sternberg cells and Hodgkin cells makes the distinction from RDD easier. Hodgkin cells and Reed–Sternberg cells are positive for CD30, CD15, dim PAX5, fascin, and MUM1, and the background histiocytes are positive for CD68 but not S100;Anaplastic large-cell lymphoma can present with a sinusoidal pattern, but the hallmark cells with horseshoe nuclei are very distinctive and diffusely positive for CD30 by IHC stains (≥75% of neoplastic cells). For the ALK-positive variant, it harbors *ALK* translocations and is positive for ALK immunostaining.ALK-positive histiocytosis is first described in 3 infants [48]. This entity is divided into three groups: Group 1A (infants with hematopoietic and liver involvement); group 1B (multisystemic diseases); and group 2 (patients with single-organ involvement [49]. Similar to RDD, ALK-positive histiocytosis shows emperipolesis and stains for histiocytic markers (CD168, CD63, CD4, CD14). Many cases also express OCT2 (61%), pERK (46%) and cyclin-D1 (49%). However, this entity harbors many ALK translocations, most commonly *KIF5B-ALK*, and stains positive for ALK1. S100 is variably expressed, unlike RDD, which is uniformly expressed [49];Histiocytic sarcoma shows histiocytic proliferation and usually has marked cytologic atypia, brisk mitotic activity, and is negative for S100. Though emperipolesis could be occasionally identified, unlike RDD, it also has an aggressive clinical course;Juvenile xanthogranuloma, most commonly present in children, with a predilection for the head-and-neck region. It is often cutaneous; however, rarely, it can be subcutaneous or intramuscular. Usually, it resolves spontaneously. Histologically, proliferation of numerous mononuclear and multinucleated cells with Touton-like features on a background of inflammatory cells (lymphocytes and eosinophils) are consistently present. There can be variable foamy histiocytes and lipids. Emperipolesis could be seen. Both diseases are positive for CD68 and CD4. However, unlike RDD, the histiocytes are negative for S100 and more frequently positive for Factor XIIIa [50];Follicular dendritic cell sarcoma may involve lymph nodes or extranodal sites. Oval-to- spindle cells with eosinophilic cytoplasm forming syncytial sheets. Tumor cells are admixed with small lymphocytes and positive for CD21, CD23, CD35, and clusterin, distinguishing it from RDD. Some cases may express S100 focally.

## 8. Genetic and Mutational Profile

Approximately 50% of cases of RDD do not have a known distinct mutational profile. From 30% to 50% of patients with RDD-harboring somatic mutations are frequently involved in *ARAF*, *NRAS*, *KRAS*, *MAP2K1*, *CSF1*, and *CBL* genes, of which *MA2P2K1* and *KRAS* were the most frequent, making up 14% and 12.5% of all RDD, respectively (see Table 2) [51,52,53,54,55] (Table 2) Garces et al. showed mutually excluded *KRAS* and *MAP21* gene mutations in RDD, together making up 33% of cases [38]. *NRAS* mutation was detected in numerous cases of purely cutaneous RDD. A study of Wu et al. showed *NRAS* (A146T) mutation was the most common among point mutants, followed by *NRAS* G13S, suggesting that the mutations may play a role in pathogenesis of cutaneous RDD [56]. *PTPN11*, *NF1* mutations have also occasionally been reported [39]. *BRAF* mutations can occasionally occur in RDD, and not just V600E mutations; other mutations were reported (*BRAF* Y472C and *BRAF* R188G, deletion in exon 12 of *BRAF*) [57,58,59]. *BRAF* (V600E) is more common in RDD overlapped with LCH or ECD diseases [60]. Given the low frequency of *BRAF* mutations in cases of RDD, one should pay an attention to RDD-overlapping diseases.

There are many other genes that are mutated in RDD, including *SNX24*, and are involved in intracellular trafficking. *INTS2*, *CIC*, *SFR1*, *BRD4*, and *PHOX2B* genes play roles in the transcriptional regulation as well as cell-cycle regulation genes (*PDS5A*, *MUC4*); ubiquitin-proteasome pathway (*USP35*); and DNA-mismatch repair genes (*BRCA1*, *LATS2*, *ATM*) [54,63].

Some gene mutations frequently identified in myeloid or lymphoid neoplasms have also been observed in RDD, e.g., *ASXL1*, *TET2*, and *DNMT3A* [39]. A large cohort of 28 patients reported many mutations, which are implicated in other myeloid malignancies, e.g., *TET2*, *MLL4*, *NF1*, *ALK*, and *ASXL1*; many of these genes are driver mutations in different myeloid and lymphoid neoplasms [59]. Whether they might play a role in the pathogenesis of RDD or if they might cause transformation or transdifferentiation to these neoplasms has yet to be determined. Many other mutations have been reported: *SEC62*, *FCGBP*, *PIK3R2*, *PIK3CA*, *VCL*, *EGFR*, *ERBB2* and *TLR8* (minimum variable allele frequency in these mutations is 2%) [59]. The exact roles of these genes are of further exploration.

Cytogenetic testing usually does not have a role in RDD; however, testing can be conducted to rule out a neoplastic process. One case was reported to have a normal karyotype with a minor clone lacking chromosome 20 [64].

Familial RDD has been reported in patients with germline mutations of *SLC29A3* [65]. Two mutations have been reported to date: p.G427S and p.G437R. The spectrum of diseases associated with a mutation in the *SLC29A3* gene mainly include the following three diseases:Familial or Faisalabad histiocytosis: Children present with sensorineural deafness and joint contractures. It is autosomal recessive disorder [65,66]. Histologically, it resembles RDD; therefore, obtaining a clinical history along with a genetic consultation is an important step;H syndrome: Children present with hypogonadism, indurated, hyperpigmented, hypertrichotic skin plaques, hepatomegaly, cardiac abnormalities, and hearing loss [67,68]. Skin lesions share histologic features with RDD;Pigmented hypertrichotic dermatosis with insulin-dependent diabetes syndrome: Children present with insulin-dependent diabetes mellitus and pigmented hypertrichosis [67].

All these diseases are described as histiocytosis-lymphadenopathy plus syndrome. A recent study confirmed that, despite sharing a missense mutation c.1088 G > A [p.Arg363Gln] of the SLC29A3 gene, the clinical phenotype of the intrafamilial SLC29A3 disorders could be heterogeneous [69].

Another cause of familial RDD is heterozygous germline mutations in the FAS and TNFRSF genes, which also cause ALPS. Patients with ALPS tend to have a male predominance, more aggressive disease behavior, and an earlier age of disease onset [70].

## 9. Prognosis

Nodal RDD has a good prognosis, which correlates with the number of nodal groups; however, this is according to an article published in 1990 [5]. To date, there has not been a large case series monitoring the prognosis of RDD patients. In extranodal RDD forms, prognosis also correlated with the number of organ systems that were involved. Prognosis is usually favorable for cutaneous, nodal, and bone RDD; however, CNS involvement with RDD has variable outcomes [13,16,71]. In most cases, it has a good prognosis; however, some patients have a progressive and fatal course. Regarding gastrointestinal involvement by RDD, a case series reported approximately 20% mortality. GU-system involvement has a worse prognosis of approximately 40% mortality in RDD involving the kidney. The largest series of RDD was reported by Foucar et al. in 1990 [5]; it showed that 17 (7%) of 238 patients died because of direct complications of their disease, infections, or amyloidosis.

## 10. Treatment

No standard therapy has been established for patients with RDD based on results of prospective clinical studies due to the rarity of this disorder. Treatment algorithms are based on retrospective case series, case reports, disease-registry analyses, and expert opinions. Most recently, the NCCN Clinical Practice Guidelines in Oncology, Histiocytic Neoplasms, Version 2.2021, proposed diagnostic and treatment algorithms for patients with RDD [23]. Observation, watch and wait*,* is considered for asymptomatic or mildly symptomatic patients as 40% of cases with nodal/cutaneous involvement will have spontaneous remission [72];Complete surgical resection is used for patients with unifocal areas of involvement. Surgical therapy can also be useful for patients with spinal cord compression or upper-airway obstruction or with large lesions which might cause end-organ damage. Regarding cutaneous RDD, surgery has been found to be the most effective line of treatment for localized disease [73]. Endoscopic resection of sinonasal RDD can help with symptomatic relief [74];External-beam radiation therapy can play a role in patients with localized unresectable symptomatic steroid-refractory masses, especially in extranodal locations. The most frequently administered doses ranged from 20 to 30 Gy with 2 Gy per fraction, but higher disease levels have also been reported. Overall response rate (ORR) was approximately 40% in patients treated with doses ranging from 30 to 49 Gy and 27% with doses <30 Gy;Corticosteroids have been used in RDD as it was found that steroids decrease the nodal size and symptoms. The optimal dose and duration for this line treatment is ambiguous; however, prednisone (40–70 mg per day) demonstrated variable responses, ranging from failure–no response to complete response in cases with bone, orbit, CNS, and autoimmune-related RDD [75]. According to the consensus recommendations published in 2018, steroids alone do not lead to a stable response among patients with extranodal RDD. It has been found that patients with RDD require a higher dose of prednisone (>0.5 mg/kg per day). Similarly, dexamethasone is effective in RDD with nodal and intracranial lesions [76,77]. A case series of 57 patients reported by Mayo Clinic group showed that corticosteroid treatment was associated with 56% ORR in treatment-naïve patients. Relapses occurred in 53% of patients. When steroids were used as a second-line therapy, the ORR was 67% [15];Sirolimus is frequently used for treating ALPS [78], which suggested a potential benefit for patients with RDD. This agent was effective in a case report of a child with resistant RDD and recurrent autoimmune cytopenias [79];Chemotherapy and immunotherapy have been used to treat RDD with many different chemotherapeutic agents, leading to a mixed treatment responses and adverse events. Some agents were ineffective, such as anthracyclines and alkylating agents; however, vinca alkaloids have shown variable results. Many different regimens have been postulated to have good responses in RDD. A case series on 15 patients with massive lymphadenopathy was reported by the University of Pennsylvania group. The patients were treated with rituximab monotherapy, resulting in 64% progression-free survival at 24 months [80]. Other chemotherapy agents used in refractory RDD with promising activities were nucleoside analogs, such as cladribine (2.1–5 mg/m^2^ per day for 5 days every 28 days for 6 months) and clofarabine (25 mg/m^2^ per day for 5 days every 28 days for 6 months) [81,82,83];Immunotherapy such as TNF-α inhibitors. thalidomide and lenalidomide were used due to the increased levels of TNF-α and interleukin-6 in patients with RDD. A case report of the patient treated with a low dose (50 mg/day) thalidomide with a dose escalation to 100 mg/day demonstrated promising results [84]. On the one hand, the results of thalidomide therapy have been mixed, and the side effects associated with administration of this agent are not negligible. Lenalidomide, on the other hand, has shown very good responses in patients with refractory nodal and bone RDD and overall better tolerance with fewer side effects, except myelosuppression compared to thalidomide [85]. Rituximab has also been used to treat autoimmune RDD; however, some cases show recurrence or become refractory [86,87,88];Targeted therapy with imatinib has been used in a case report with refractory RDD. Imatinib could be an option for systemic involvement by non-LCH disorders [89]. Interestingly, the lesional histiocytes were positive for expression of PDGFRB and KIT/CD117 by immunohistochemical stain, but no underlying mutations in genes coding for these tyrosine kinases were identified. *BRAF* mutations have rarely been observed in RDD. BRAF inhibitor (dabrafenib) was reported to be used in a patient with concurrent RDD and LCH with clinical and radiological response but increased *BRAF*-V600E post 13-month therapy; additional MEK inhibitor (trametinib) was added with limited follow-up time [60]. MEK inhibitor (cobimetinib) has shown good results in a patient with *KRAS* (p.G12R) mutated RDD [53]. Moyon et al. reported that two RDD patients received cobimetinib showed significant pulmonary response with regard to metabolism and tumor size [19].

## 11. Conclusions

In the past, RDD was thought to be a benign histiocytic proliferative disorder of unknown etiopathogenesis. Most recently, based on molecular and genetic data, a consensus has been reached defining RDD as a neoplastic myeloproliferative process with causative mutations in the MAPK pathway in a subset of cases. Additional IHC markers have been found to help refine the diagnosis of RDD, such as cyclin-D1 and OCT2. Even though standard therapy based on prospective clinical trials has not been well established to date, the discovery of novel driver mutations could be useful for the development of novel targeted therapy, resulting in better tolerance and outcomes.

## Figures and Tables

**Figure 1 cancers-14-05271-f001:**
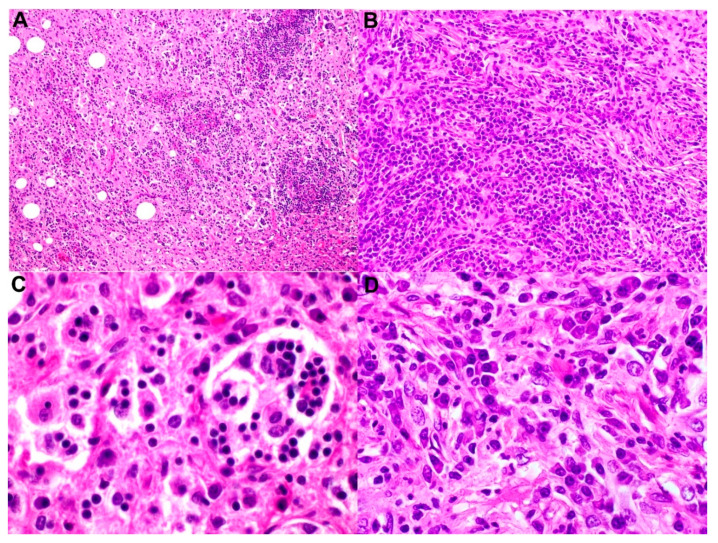
The selected images show a case of nodal Rosai–Dorfman disease with concurrent IgG4-related disease. (**A**) Low-power magnification shows effacement of normal lymph-node architecture by histiocytes (H & E, magnification 40×). (**B**) Higher-power magnification demonstrates increased plasma cells arranged in nests and singly associated with background fibrosis and histiocytes (immunoperoxidase, magnification 200×). (**C**) Loaded histiocytes with emperipolesis and (**D**) an increased number of plasma cells (immunoperoxidase, magnification 600× and 600×, respectively).

**Figure 2 cancers-14-05271-f002:**
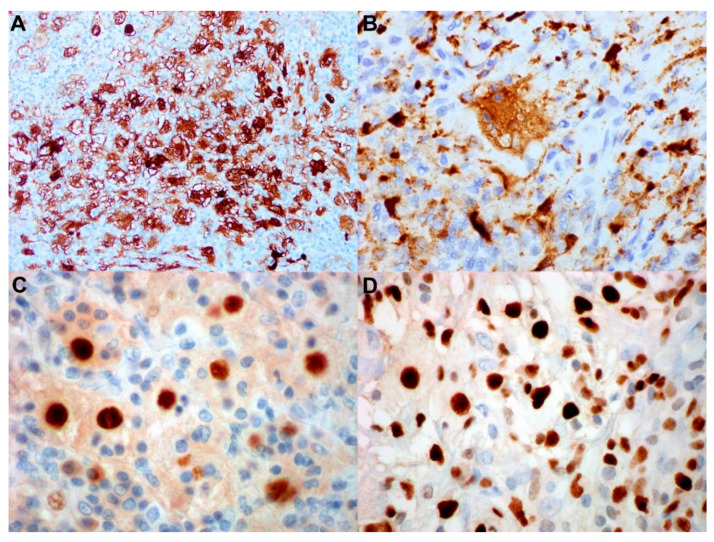
Routine immunohistochemical stains for Rosai–Dorfman disease. (**A**) S-100 immunohistochemical stain highlighting the histiocytes, which also demonstrates emperipolesis (immunoperoxidase, magnification 600×). (**B**) CD68 immunohistochemical stain highlighting the histiocytes (immunoperoxidase, magnification 600×). (**C**,**D**) BCL-1 and Oct2 immunohistochemical stains are positive for histiocytes of Rosai–Dorfman disease (immunoperoxidase, magnification 600×).

**Figure 3 cancers-14-05271-f003:**
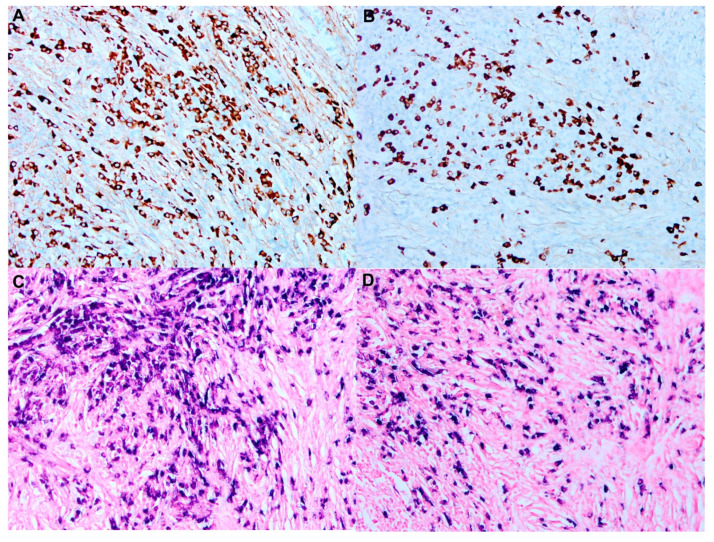
Representative images of IgG4 disease in a patient with Rosai–Dorfman disease (refer to Figure 1): increased plasma cells in a background of fibrosis. (**A**,**B**) Immunohistochemical stains demonstrate plasma cells positive for IgG and proportionally positive forIgG4 (**A**,**B**). immunoperoxidase 100×, respectively). (**C**,**D**) In situ hybridization using kappa and lambda light chain probes reveal the plasma cells to be polyclonal (in situ hybridization 100×, respectively).

**Table 1 cancers-14-05271-t001:** Clinical Symptoms and Signs of Nodal and Extranodal RDD—Organ and Tissue Specified [6,15,16,18,19,20,21,22].

**Site**	**Incidence**	**Symptoms/Signs**	**Radiologic Findings**
**Nodal**			
Lymph node	57% of cases	Bilateral cervical lymphadenopathy or other LN sites, manifested with palpable masses	Enlarged LNs
**Extranodal**			
Skin	10% of cases	Painless, macular, slow-growing papules, subcutaneous nodules. Any skin site can be affected.	Nodule(s) or mass(es)
CNS	<5% of cases (75% intracranial and 25% spinal lesions), more than 300 cases have been reported. Cervical and thoracic regions are the most common areas affected in spinal-dural or epidural lesion	Headaches, seizures, gait difficulty. In familial cases, there is an association with damage to the auditory nerve pathway and deafness.	Dural lesion, extra-axial, homogeneously enhancing, mimicking nodular meningioma; or parenchymal (infratentorial) involvement
Orbit	11% of cases	Presents as a mass in different part of the orbit, e.g., conjunctiva, lacrimal glands, and cornea. It can also present as uveitis.	Orbital mass
Head and neck	11% of cases involving nasal cavity, more common among Asians	Nasal obstruction, epistaxis, and nasal dorsum deformity	Nodules, swelling, mass(es)
Intrathoracic	2% of patients, with pulmonary disease concurrent lymphadenopathy or systemic disease. Cardiac involvement is extremely rare ~0.1–0.2% of cases.	Chronic dry cough, progressive dyspnea, or acute respiratory failure. RDD affecting lower respiratory tract have a high mortality rate, reaching 45%.	Pulmonary nodular consolidation in all lobes of the lungs; pleural effusion with fibrosis or nodules
Retroperitoneal /genitourinary tract	Kidneys are affected in approximately 4% of cases.	Abdominal or flank pain, fullness, hematuria, renal failure, hypercalcemia, and/or nephrotic syndrome	Mass(es), hydronephrosis, urethral obstruction
Gastrointestinal tract	<1% of cases, commonly in middle-aged women with concurrent nodal or extranodal affection	Abdominal pain, constipation, hematochezia, and intestinal obstruction	Mass(es)
Bone	5% to 10% of cases, usually with concurrent nodal affection	Bone pain and, rarely, pathologic fractures	Cortex-based osteolytic lesion, commonly long bones, vertebrae, and sacrum

Abbreviations: CNS, central nervous system; RDD, Rosai–Dorfman disease. LN: lymph node

**Table 2 cancers-14-05271-t002:** Certain gene mutations identified in RDD [6,38,55].

Gene	Molecular Alteration	References
*ARAF*	N217K	[51]
*MAP2K1*	F53V, L115V, P124R, G128D, V50M, D65M	[38,51,52]
*KRAS*	A146T, A146V, K117N, G12D, G12R, G13S, Q22K,	[38,51,53,55,56,61]
*NRAS*	G13D	[51]
*CBL*	C384Y, GNAQ Q209H	[55]
*KDM5A*	amplification	[55]
*FBXW7*	E113D	[55]
*BRAF*	V600E, deletion (p. 486–491), Y472C and R188G	[57,58,59]
*SMAD4*	T521I (variant of unknown significance, 1 case)	[62]

## Data Availability

Not applicable.

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
