# Peer review of "Rosai–Dorfman Disease between Proliferation and Neoplasia"

_cancers, 2022, doi:10.3390/cancers14215271_

Round 1

Reviewer 1 Report

The authors provide a review on Rosai-Dorfman disease. It is well written and well documented, but should be improved.

P5 “RDD with concurrent disorders”: mixed RDD-LCH is over-developped (only cases reports), while RDD-Erdheim-Chester disease with is more frequent should be more detailed.

P5 “histopathology” Description of “unique feature” should include the large nuclei with pale chromatin and large nucleoli of the large histiocytes with emporipolesis.

P7 “differential diagnosis”. Too long for some diseases, while ALK+ histiocytosis, which is frequently S100+ and OCT2+ is missing

P9. I’m not English-native, but google is, and none of us knows what “prognostiation” means. This chapter should mainly start and insist that the level of evidence are very low. The main paper on prognosis was published 32 years ago and based on retrospective declarative data.

Table 2 should have major changes (or be deleted).  Should include all genes activating the MAPkinase pathway, and not the others.  One gene/line and all published pathogenic variants of one gene in the same line. Provide all references or none.

Figure 1 should be replaced by good figures. The “lymph node” looks like skin biopsy! (see upper left corner of A). The large cells of C and D are not typical of RDD.

References: 26/72 (37%) are older than 2010. The authors should select more recent

Several typo should be corrected in text, including with affiliations and legends of figures

Author Response

P5 “RDD with concurrent disorders”: mixed RDD-LCH is over-developed (only cases reports), while RDD-Erdheim-Chester disease with is more frequent should be more detailed.

  • We totally agree with your point, we have actually added in the concurrent diseases another paragraph talking about ERD overlapping with RDD

P5 “histopathology” Description of “unique feature” should include the large nuclei with pale chromatin and large nucleoli of the large histiocytes with emperipolesis.

  • We have added more details in the histology which included the point you made
  •  

P7 “differential diagnosis”. Too long for some diseases, while ALK+ histiocytosis, which is frequently S100+ and OCT2+ is missing

  • Based on your suggestion, we have added ALK+ histiocytosis, and made differential diagnoses more concise, and separate them into nodal and extranodal base diseases.
  •  

P9. I’m not English-native, but google is, and none of us knows what “prognostication” means. This chapter should mainly start and insist that the level of evidence are very low. The main paper on prognosis was published 32 years ago and based on retrospective declarative data.

  • We made a change from "prognostication" to "prognosis". We admitted the paper regarding prognosis published 32 years ago was too old. However, no updated information is available per literature searching. We recommend large study to reanalyze the prognostic factors and clinical outcomes based on updated molecular profile and treatment strategies.
  •  

Table 2 should have major changes (or be deleted).  Should include all genes activating the MAPkinase pathway, and not the others.  One gene/line and all published pathogenic variants of one gene in the same line. Provide all references or none.

  • We revised the table 2 according to your suggestion and add the corresponding references and separated mutations involved MAPK pathway and other genes.
  •  

Figure 1 should be replaced by good figures. The “lymph node” looks like skin biopsy! (see upper left corner of A). The large cells of C and D are not typical of RDD.

  • We have made Figure 1 which included more representative images of RDD histology. Figure 2 has also been updated by replacement with cyclin D1/BCL-1 and Oct 2 IHC images.

References: 26/72 (37%) are older than 2010. The authors should select more recent

  • According to your critiques and suggestion, we have gone through all references and include more updated ones published recently (2015-2022).
  •  

Several typo should be corrected in text, including with affiliations and legends of figures

  • Yes, sorry for a rash version without checking carefully. We have asked our editorial office at Moffitt Cancer Center to make significant English proof-reading and edition.

Reviewer 2 Report

The authors report a review about Rosai-Dorfman disease, a rare histiocytosis now considered a myeloid neoplasm. 

This work reports the landscape of RDD from historical presentation to overlapping with other conditions making the diagnosis difficult. 

The pathological description is very clear and evokes the difficulty of differential diagnosis.

Despite the quality of the work, there is some point to consider for better clarity. 

The authors say that it is not uncommon to accompany another histiocytosis. This statement is difficult to share as only two case reports mentioned overlap between RDD and LCH. For ECD/RDD overlap, the biggest series from 3 referral centers describes 13 patients. 

I think that it would be interesting  for clinicians to have a better description of the clinical and radiological pattern of the disease, especially for CNS and pulmonary involvement ( 10.1093/neuonc/noab107 and 10.1016/j.chest.2019.09.036)

It would also be of interest  to describe features associated with associated disease ( SLE, Ig-G4 related disease, ECD)

Regarding therapies, targeted therapies ( MEK-inhibition) have to be considered in patients with life-threatening conditions, especially in phosphor-ERK positivity on tissue biopsy. Furthermore, Rituximab is also reported as an efficacious treatment for ITP-associated RDD.

The authors should also revise the manuscript for orthograph.

Author Response

The authors say that it is not uncommon to accompany another histiocytosis. This statement is difficult to share as only two case reports mentioned overlap between RDD and LCH. For ECD/RDD overlap, the biggest series from 3 referral centers describes 13 patients. 

  • We do agree with you, we have changed the wording to avoid confusion and added reference to this manuscript of ECD and RDD overlapping diseases.
  •  
  • I think that it would be interesting for clinicians to have a better description of the clinical and radiological pattern of the disease, especially for CNS and pulmonary involvement (10.1093/neuonc/noab107 and 10.1016/j.chest.2019.09.036)

  • We have added the references you pointed and included clinical and radiological pattern of RDD with CNS and pulmonary involvement in Table 2.

  • It would also be of interest to describe features associated with associated disease (SLE, Ig-G4 related disease, ECD)
  • We have added SLE, ECD and its overlap with RDD. Ig-G4 related disease with RDD is updated.

  • Regarding therapies, targeted therapies (MEK-inhibition) have to be considered in patients with life-threatening conditions, especially in phosphor-ERK positivity on tissue biopsy. Furthermore, Rituximab is also reported as an efficacious treatment for ITP-associated RDD.

  • Our clinician Dr. Sokol agreed with your points. Rituximab treatment for RDD, especially autoimmune related situation, is added in treatment part.
  •  
  • The authors should also revise the manuscript for orthograph.

  • We have made it according to your suggestion.

Reviewer 3 Report

The manuscript by Elbaz Younes et al is a comprehensive review of Rosai-Dorfman Disease encompassing the recent advancements in literature. Overall, the manuscript is a valuable addition with summary of the mutations in RDD. The figures are nicely presented.

The details can be refined further, and I suggest the following:

-     1. Given that this is a review, the authors need to mention about RDD classification under the ‘C’ group (cutaneous RDD) defined by the Histiocyte Society (PMID: 26966089).

-      2. RDD is primarily a histopathologic diagnosis and does not always require clinical and radiologic features to be present unlike ECD. Under the abstract discussion section, I suggest rephrasing the sentence “RDD is a complex diagnosis that needs clinical, radiologic, and pathologic features to be present”

-      3. Under subclassification section, please correct the error in phrasing of the sentence: it should include NLPHL and follicular lymphoma:  The most common two types of lymphomas associated with RDD are nodular lymphocyte predominant Hodgkin lymphoma. The name of this lymphoid malignancy is now changed to "nodular lymphocyte predominant B-cell lymphoma"according to the most recent International Consensus Classification of Mature Lymphoid Neoplasms [11]. and follicular lymphoma.

-      4. For the figure 2: as you mention about CyclinD1 and OCT2 expression in the review, please add representative images for the same.

-      5. Pages 6-7: The section on differential diagnoses should be formatted. Separate the differential diagnoses into benign and malignant entities and describe how each differs from RDD in terms of histopathology. Add Sinus histiocytosis as a separate differential diagnosis along with benign entities described. Add ECD to the differential diagnosis and explain how the clinicopathologic and radiologic findings is critical for diagnosis.

-      6. SLC29A3 mutations is detailed under Familial RDD in the discussion; please exclude this from the differential diagnosis section

-      7. Under genetics section, although the BRAF V600E is usually negative in RDD, it can be present in mixed histiocytosis (overlap of RDD and LCH; PMID: 31213430). A rare case of BRAF V600E mutated RDD has been previously described (PMID: 29748446). The authors need to address these in their review and possibly add a comment that the presence of BRAF V600E in an otherwise classic histopathology of RDD should prompt clinical and radiologic evaluation for mixed histiocytosis (ECD/RDD or LCH/RDD overlap).

-     8. The authors have discussed about overlap of RDD and LCH but the details regarding overlap of RDD and ECD (PMID: 31123032) could be expanded upon under the section RDD with concurrent disorders.

Author Response

 Given that this is a review, the authors need to mention about RDD classification under the ‘C’ group (cutaneous RDD) defined by the Histiocyte Society (PMID: 26966089).

  • You are correct. We added the classification and specifically said that cutaneous RDD is under the “C” group.

-      2. RDD is primarily a histopathologic diagnosis and does not always require clinical and radiologic features to be present unlike ECD. Under the abstract discussion section, I suggest rephrasing the sentence “RDD is a complex diagnosis that needs clinical, radiologic, and pathologic features to be present”

  • We have changed it to be clearer to our readers.

-      3. Under subclassification section, please correct the error in phrasing of the sentence: it should include NLPHL and follicular lymphoma:  “The most common two types of lymphomas associated with RDD are nodular lymphocyte predominant Hodgkin lymphoma. The name of this lymphoid malignancy is now changed to "nodular lymphocyte predominant B-cell lymphoma"according to the most recent International Consensus Classification of Mature Lymphoid Neoplasms [11]. and follicular lymphoma.”

  • According to ICC, it has changed to nodular lymphocyte predominant B-cell lymphoma, however, the WHO edition still uses this name. We keep the NLPHL and new term used by ICC as many readers might not be familiar with the change.

-      4. For the figure 2: as you mention about CyclinD1 and OCT2 expression in the review, please add representative images for the same.

  • Figure 2 are modified by replacement with cyclin D1/BCL1 and Oct2 IHC images.

-      5. Pages 6-7: The section on differential diagnoses should be formatted. Separate the differential diagnoses into benign and malignant entities and describe how each differs from RDD in terms of histopathology. Add Sinus histiocytosis as a separate differential diagnosis along with benign entities described. Add ECD to the differential diagnosis and explain how the clinicopathologic and radiologic findings is critical for diagnosis.

  • We divided the differential diagnosis into benign and malignant as suggested. We also added ECD to the differential as suggested with the relevant clinicopathologic and radiologic differences

-      6. SLC29A3 mutations is detailed under Familial RDD in the discussion; please exclude this from the differential diagnosis section

  • We deleted it from the differential diagnoses as suggested

  1. Under genetics section, although the BRAF V600E is usually negative in RDD, it can be present in mixed histiocytosis (overlap of RDD and LCH; PMID: 31213430). A rare case of BRAF V600E mutated RDD has been previously described (PMID: 29748446). The authors need to address these in their review and possibly add a comment that the presence of BRAF V600E in an otherwise classic histopathology of RDD should prompt clinical and radiologic evaluation for mixed histiocytosis (ECD/RDD or LCH/RDD overlap).

  • Yes, we should discuss about BRAF mutations in RDD, thought rare. We have added the case reports of RDD with BRAF mutations as well as the cases where overlap between LCH and RDD with BRAF V600E mutation. Given the low frequency of BRAF mutations in cases of RDD, one should prompt clinical and radiologic assessment for composite or concurrent mixed histiocytic neoplasm (RDD/ECD or RDD/LCH) when presence of BRAF V600E mutation in an otherwise classic histopathology of RDD.
  •  

-     8. The authors have discussed about overlap of RDD and LCH but the details regarding overlap of RDD and ECD (PMID: 31123032) could be expanded upon under the section RDD with concurrent disorders.

  • We have expanded on those entities as suggested. Recommended citations are included in the revised manuscript.

Round 2

Reviewer 1 Report

the authors responded to comments.

I still see "11. Prognostication". please change in prognosis

some edits also (LCD instead of LCH among examples)

Author Response

Both have been changed as required 
